# Short-Term UVB Irradiation Leads to Persistent DNA Damage in Limbal Epithelial Stem Cells, Partially Reversed by DNA Repairing Enzymes

**DOI:** 10.3390/biology12020265

**Published:** 2023-02-07

**Authors:** Thomas Volatier, Björn Schumacher, Berbang Meshko, Karina Hadrian, Claus Cursiefen, Maria Notara

**Affiliations:** 1Department of Ophthalmology, Faculty of Medicine and University Hospital Cologne, University of Cologne, 62, 50937 Cologne, Germany; 2Cologne Excellence Cluster for Cellular Stress Responses, Aging-Associated Diseases (CECAD) and Center for Molecular Medicine (CMMC), University of Cologne, Joseph-Stelzmann-Strasse 26, 50931 Cologne, Germany; 3Institute for Genome Stability in Aging and Disease, Faculty of Medicine and University Hospital Cologne, University of Cologne, 50931 Cologne, Germany; 4Center for Molecular Medicine Cologne (CMMC), Faculty of Medicine and University Hospital Cologne, University of Cologne, 21, 50931 Cologne, Germany

**Keywords:** cornea, UV, DNA damage, limbal stem cells, proteomics

## Abstract

**Simple Summary:**

Ultraviolet light from the sun causes DNA damage and is a major exogenous genotoxin, particularly affecting the skin and eyes of humans. UV-induced lesions are repaired by complex DNA repair mechanisms that sometimes fail, leading to burns or cancers. Here, we probe the potential of two ultraviolet damage-specific repair enzymes not found in humans: one from the kangaroo rat and one from an anti-bacterial virus. Both can be produced with relatively low costs. While these two repair enzymes have been studied in the skin and some products are already commercially available, comparatively little research has been conducted for the eyes, and there are no commercially available products. Therefore, we aim to offer new options for the protection of eyes that are particularly sensitive to ultraviolet rays and require more rapid repair than normal.

**Abstract:**

The cornea is frequently exposed to ultraviolet (UV) radiation and absorbs a portion of this radiation. UVB in particular is absorbed by the cornea and will principally damage the topmost layer of the cornea, the epithelium. Epidemiological research shows that the UV damage of DNA is a contributing factor to corneal diseases such as pterygium. There are two main DNA photolesions of UV: cyclobutane pyrimidine dimers (CPDs) and pyrimidine-pyrimidone (6–4) photoproducts (6-4PPs). Both involve the abnormal linking of adjacent pyrimide bases. In particular, CPD lesions, which account for the vast majority of UV-induced lesions, are inefficiently repaired by nucleotide excision repair (NER) and are thus mutagenic and linked to cancer development in humans. Here, we apply two exogenous enzymes: CPD photolyase (CPDPL) and T4 endonuclease V (T4N5). The efficacy of these enzymes was assayed by the proteomic and immunofluorescence measurements of UVB-induced CPDs before and after treatment. The results showed that CPDs can be rapidly repaired by T4N5 in cell cultures. The usage of CPDPL and T4N5 in ex vivo eyes revealed that CPD lesions persist in the corneal limbus. The proteomic analysis of the T4N5-treated cells shows increases in the components of the angiogenic and inflammatory systems. We conclude that T4N5 and CPDPL show great promise in the treatment of CPD lesions, but the complete clearance of CPDs from the limbus remains a challenge.

## 1. Introduction

The corneal epithelium is composed of five to six layers of non-cornified squamous cells. The basal layer comprises tightly packed cuboidal cells that adhere to the Bowman membrane [1]. The corneal epithelium is a high turnover barrier that produces cytokines to influence neighboring cells, primarily corneal keratocytes [2]. Basal epithelial cells are intermediate progenitors maintained by, and descended from, limbal epithelial stem cells [1]. Within the limbus are the palisades of Vogt; limbal stem cells reside between these ridges [3]. The limbus is the interface between the vascularized conjunctiva and the avascular cornea. The palisades are partially vascularized. The vessels provide nutrients and oxygen to the limbal cells [4]. The limbus and its cells maintain the conjunctiva–cornea border. Limbal damage enables the invasion of conjunctival cells into the cornea. This leads to vascularization and corneal opacification [5,6,7]. Degradation of the limbus may occur following UV damage, where the limbal cells can no longer maintain the border [8,9,10,11]. The role of epithelial factors in the maintenance of corneal angiogenic and lymphangiogenic privilege is evident; the role of the limbus as a physical barrier is not as well researched [12,13]. Rapid repair of any damage is important to prevent microbial intrusion or further trauma [14]. The replacement of the epithelial cell population is necessary for wound closure and new epithelial cells descend from the limbal stem cells. The healing of epithelial wounds can be delayed or even interrupted by any damage to the limbus [15].

UV radiation causes lesions in cellular DNA. These lesions come in the form of either cyclobutane pyrimidine dimers (CPDs) or pyrimidine (6-4) pyrimidone photoproducts (6-4PPs) [16]. CPD and 6-4PP lesions both occur when two adjacent pyrimidines bind to each other covalently [17,18,19]. CPDs occur more frequently than 6-4PPs, with CPD lesions estimated to occur three times as frequently as 6-4PP lesions. The cells can either repair or can perform preventative apoptosis. If both the repair and apoptotic processes fail, the descendant cells of the damaged parent may carry mutations [20,21,22]. Due to their greater steric aberration in the DNA, 6-4PPs are more effectively recognized, and thus repaired, by NER than CPD lesions [23]. The DNA repair enzymes, CPD photolyase (CPDPL) and 6-4PP-photolyase, bind to either a CPD or a 6-4PP, respectively. These cryptochrome enzymes absorb visible (blue) light as an energy source for electron transfer. This electron transfer allows the enzyme to break the covalent bond that was formed between the pyrimidines [24,25,26].

Photolyases come in various forms and are found in several organisms, ranging from virus to fungi and marsupials. Humans lack CPD and 6-4PP photolyases and must rely on nucleotide excision repair (NER) pathways to remove photolesions [27,28]. The NER pathway is not a single specialized enzyme, it is a long cascade of detection and repair proteins. This allows the NER to repair several kinds of DNA damage rather than just CPD and 6-4PP. NER pathways are less effective in recognizing CPD and 6-4PP lesions than their respective photolyases. NER proteins will not target UV damage specifically, like the photolyases, but they are capable of repairing CPDs and 6-4PPs [29,30]. NER comes in two main forms: global genome (GG-) NER and transcription-coupled (TC-) NER. TC-NER and GG-NER differ mainly in the way they recognize DNA damage. GG-NER relies on the XPC complex constantly probing the DNA for lesions, while TC-NER relies on RNA polymerase stalling when it encounters damaged DNA [31]. Mutations of the genes encoding for NER proteins often result in UV hypersensitivity of the skin.

The skin and the cornea face similar UV challenges but respond differently. Ocular surface pathologies such as pterygium, intraepithelial neoplasia, or carcinoma, have a lower rate of incidence than their equivalent in the skin [32,33,34]. There is a clear correlation between UV exposure and the occurrence of ocular surface pathologies, with high incidences of UV-related conditions within 30 degrees latitude of the equator, where the UV radiation is high [34,35]. Lesions occur more often in the sun-exposed interpalpebral fissure, specifically in the nasal or temporal regions within the limbus [36]. The eyes are one of the areas most exposed to UV light, with an estimated 5 to 10% of all skin cancers occurring in the eyelids [37]. UV does drive mutation and cancerous lesions in the cornea, but the corneal epithelia is less prone to UV-induced cancer than skin [38].

T4 endonuclease V (T4N5), encoded by the bacteriophage T4 endo V, is a repair enzyme that can target CPDs [39]. Purified T4N5 can be used as a CPD repair option in vulnerable tissue, but the enzyme needs to be delivered to the nucleus to access DNA. Liposome delivery system were used with T4N5, with initial delivery into cell cultures and applications to skin [40,41]. With CPD repair, T4N5 can perform the same function as the multi-complex NER mechanisms. In situations where the DNA repair mechanisms are faulty such as in XP, T4N5 can have an even greater impact and can prevent the development of carcinomas [42]. In skin, liposomal T4N5 delivery has had beneficial effects on CPD repair [43]. This has led to the design and distribution of T4N5-containing pharmaceutical products as novel preventative options for particularly photo-vulnerable patients [44]. The use of photolyase in such products presents a possible continuation of the effort to include not just photo-blocking measures but also DNA-repairing elements in UV screens. This has been the focus of several investigations, and there have already been several medical and cosmetic products developed that incorporate various proprietary photolyase blends [45,46,47,48,49]. CPD photolyase extracted from blue-green algae has been adapted to treatments for human skin [50,51]. Importantly, none of these treatments have been applied to the same extent in the cornea; there are no commercially available products, and there is no currently published research that comprehensively addresses the need for photolesion repair in the cornea. T4N5 and photolyase eye drops could be applied to the cornea to repair photolesions or to protect particularly vulnerable tissue, such as recent transplants or that of NER-deficient patients. As with any treatment, and particularly with treatments that affect DNA, there are concerns about toxicity and side-effects. In the case of T4N5, preclinical toxicological tests with single oral doses showed no negative effects [52,53]. Testing on human cells reported that T4N5 had a half-life of 3 h [54]. Further testing on animal eyes with repeated topical application to the cornea found that T4N5 was not an ocular irritant to rabbits or mice and caused no observable histopathological changes.

As there are no commercially available treatments that use T4N5 or photolyases to treat cornea photolesions, we aim to investigate the applications of these enzymes to the eye surface. The potential of both enzymes for corneal application is clear due to previous research and successful application in skin. We would use established treatments to build novel solutions to corneal UV damage repair, with particular interest in protecting the corneal limbus.

## 2. Materials and Methods

### 2.1. Primary Human Limbal Epithelial (HLE) Cell Culture

The corneal limbal epithelial cells were isolated from human corneo-scleral rims and corneal buttons. These were leftover tissue from surgery. Ethics approval (State of Cologne Ethics Approval Committee, decision number 15-093), as well as informed consent from the families of the tissue donors, was obtained according to the Declaration of Helsinki.

The tissue was transported in transport buffer (phosphate-buffered saline, 10% fetal bovine serum, 1% antibiotic/antimycotic) and then transferred to 2 mL of 1.2 U/mL dispase II solution (Sigma, Munich, Germany) overnight at 4 °C or for 2 h at 37 °C. Following digestion, the tissue was placed into a single well of a 6-well plate. The epithelial cells were gently removed by scraping using a scalpel and aiming at the limbus to isolate an enriched stem cell population. Once the cornea had been thoroughly scraped, the tissue was removed and CnT5.7 culture media (CellNTec, Bern, Switzerland) was added to the well.

Once the cells had density in the well and had reached a confluence of approximately 80%, the cells were washed with PBS and detached with TrypLE. The detached cells were resuspended in serum-containing media and passed through a cell strainer. The strained cell suspension was then centrifuged, and the pellet was resuspended in 80 µL of MACS buffer (PBS, 5% BSA, 0.4% EDTA). The cells were then incubated with anti-fibroblast magnetic beads (Miltenyi Biotec, Bergisch Gladbach, Germany) for 30 min and passed through a pre-moistened magnetic filter column mounted in its magnet stand (Miltenyi Biotec, Germany). The filtered cells were then centrifuged, resuspended, and seeded at an appropriate amount for the cell culture.

### 2.2. UVB Irradiation and T4 Endonuclease Treatment

Human limbal epithelial (HLE) cells were seeded at a density of 2 × 10^4^ cells per cm^2^ in either 12-well plates or 8-well Labtek chamber slides (ThermoFischer, Waltham, MA, USA). The cells were allowed to adhere overnight in CnT5.7 media. In a BioPorter tube (Sigma Aldrich), 20 μL of T4 endonuclease stock at a concentration of 10,000 units/mL (New England Biolabs, Ipswich, MA, USA) was mixed with 20 μL of PBS. The mix was vortexed briefly and then incubated overnight at 4 °C. After the incubation, 360 μL of Imaging solution (Gibco) was added to the BioPorter tube for a T4N5 concentration of 500 units/mL. The tube was vortexed. The media on the cells was replaced with a thin layer of PBS. The cells were irradiated with 0.03 J/cm^2^ UVB using a BioSun machine (Vilber Lourmat, Collégien, France). The UVB intensity was measured by the machine’s internal sensor. The PBS on the cells was removed, and 200 μL of the BioPorter/T4N5/Imaging solution mixture was added to the well (for the chamber slides, 100 μL was added per well). For a vehicle control, the T4N5 was replaced with an equivalent volume of PBS. The untreated control received only Imaging solution. After 3 h of incubation at 37 °C, the wells were topped up to 1 mL with CnT5.7 culture media and returned to 37 °C incubation for up to 96 h (the chamber slides were topped up to 300 μL per well). The action time of T4N5 was decided based on the manufacturer’s instructions for the BioPorter and pilot experiments with T4N5.

For the fluorescent imaging, the chamber slide wells were rinsed with PBS once before fixing the cells in 4% PFA for 10 min. For proteomics, the cells were detached with TrypLE and centrifuged at 500× *g* for 5 min. The cells were resuspended in PBS and centrifuged again to wash the extracellular material away. This washing procedure was performed twice, and the resulting pellet was snap frozen in liquid nitrogen.

### 2.3. Harvesting and Irradiating Mouse Eyes

All harvesting of animal tissue was approved by the Landesamt für Natur, Umwelt und Verbraucherschutz Nordrhein-Westfalen and conducted in strict adherence to the guidelines listed by the Association for Research in Vision and Ophthalmology Statement for the Use of Animals in Ophthalmic and Vision Research. The mice were housed under standard conditions with a 12:12 h light–dark cycle and ad libitum access to food and water.

The mouse line used in this study was the transgenic hemizygous C57BL/6 J mice expressing the Potorous tridactylus CPD photolyase (CPDPL) under the control of the Cytokeratin 14 (K14) promoter (K14-CPD-PL; (Jans et al., 2006; Schul et al., 2002)). The animals used were between the ages of 12 weeks and 36 weeks. The mice were sacrificed via cervical dislocation and whole eyes were harvested. The eyes were fixed onto a spike, submerged in PBS, and irradiated with 1 J/cm^2^ UVB. Photoreactivation took place immediately afterwards using white LED light for 30 min.

The whole eyes were then submerged in 4% PFA for 4 h at room temperature. The eyes were washed with PBS and then submerged in 20% sucrose overnight at 4 °C. The cornea was separated from the rest of the eye, cleaned, and flattened for staining. For sections, the whole eye was mounted in OCT media and cryo-sectioned.

### 2.4. Immunohistochemistry: CPD Staining in Mouse Cornea and Human Epithelial Cells

Both the mouse corneas and the human cells had their DNA denatured with 2 M HCl for 30 min. The samples were then washed several times to ensure no acid remained. The human cells were blocked for 1 h in 2% BSA in PBST. The mouse corneas were blocked with the following sequence of blocking solutions: 1 h in 2% BSA in PBST; 10 min in Normal Serum Block (BioLegend, San Diego, CA, USA); 15 min in Avidin solution (BioLegend, USA); 15 min in Biotin solution (BioLegend, USA); and 1 h in mouse-on-mouse (MOM) kit IgG blocking reagent (Vector Laboratories, Burlingame, CA, USA). After blocking, the samples were washed with PBS and the primary antibodies diluted in MOM kit working solution were applied to the cornea: mouse anti-CPD antibody (CosmoBio, San Diego, CA, USA), rabbit anti-p63a antibody (Abcam, Cambridge, UK), and rabbit anti-Cytokeratin 14 antibody. The antibodies were left to incubate overnight at 4 °C. The human cells were stained with the same antibody combination diluted in 2% BSA. The samples were washed, and the secondary antibodies were applied: MOM biotinylated anti-mouse IgG (Vector Laboratories, USA) and anti-rabbit Alexafluor488 (ThermoFisher, USA) diluted in 2% BSA. The secondary antibodies were left to incubate for 1 h at room temperature in the dark. After washing, the samples were incubated with streptavidin-Alexafluor555 diluted in 2% BSA for 10 min. The samples were counterstained with DAPI, washed, and mounted. Images were taken with an Olympus BX63 microscope and analyzed with FIJI. All the images used for the intensity measurements were taken with the same settings on the same day. For the mouse cornea sections, three images were taken for each biological repeat. These technical repeats imaged the limbus as well as the center of the cornea.

### 2.5. Proteomics

#### 2.5.1. Labelling for SP3

For the quantitative proteomics, the harvested cell pellets were dissolved in 5% SDS in 1x PBS, and the nucleic acid in the samples was degraded with Benzonase HC (25 Units per 5 × 10^5^ cells). Next, Dithiothreitol (DTT) to a concentration of 5 mM was added, and the samples were incubated for 30 min at 55 °C. Then, Chloroacetamide (CAA) to a concentration of 40 mM was added and the samples were left to incubate in the dark at room temperature for 30 min. The protein yields were estimated with A280 method measurements performed on a Nanodrop 2000c. Samples from two treatment groups were selected for analysis: UV-irradiated HLE, as controls, and the UV-irradiated HLE that received the T4 endonuclease V treatment.

The samples were analyzed on a Q Exactive Exploris 480 (Thermo Scientific) mass spectrometer equipped with a FAIMSpro differential ion mobility device (Thermo Scientific) that was coupled to an UltiMate 3000 HPLC (Thermo Scientific). The samples were loaded onto a 5 µm PepMap trap cartridge precolumn (Thermo Scientific) and reverse-flushed onto an in-house packed analytical pulled-tip column (30 cm–75 µm I.D., filled with 2.7 µm Poroshell EC120 C18, Agilent, Santa Clara, CA, USA). The peptides were chromatographically separated at a constant flow rate of 300 nL/min and the following gradient: initial 2% B (0.1% formic acid in 80% acetonitrile), up to 6% in 1 min, up to 32% B in 72 min, up to 55% B within 7.0 min, and up to 95% solvent B within 2.0 min, followed by a 6 min column wash with 95% solvent B. The FAIMS pro was operated at −50 compensation voltage and electrode temperatures of 99.5 °C for the inner and 85 °C for the outer electrode.

#### 2.5.2. Quantitation

The MS1 scans were acquired from 390 *m*/*z* to 1010 *m*/*z* at 15 k resolution. The maximum injection time was set to 22 ms and the AGC target to 100%. The MS2 scans ranged from 300 *m*/*z* to 1500 *m*/*z* and were acquired at 15 k resolution with a maximum injection time of 22 msec and an AGC target of 100%. DIA scans covering the precursor range from 400 to 1000 *m*/*z* and were acquired in 75 × 8 *m*/*z* staggered windows, resulting in 150 nominal 4 *m*/*z* windows after demultiplexing. All the scans were stored as centroids.

#### 2.5.3. Data Processing

Thermo raw files were demultiplexed and transformed to mzML files using the msconvert module in Proteowizard. A Human canonical Swissprot fasta file (downloaded 26 June 2020) was converted to a Prosit upload file with the convert tool in encyclopedia 0.9.0 (Searle 2018) using the default settings: Trypsin, up to 1 missed cleavage, range 396 *m/z*–1004 *m/z*, charge states 2+ and 3+, default charge state 3, and NCE 33. The csv file was uploaded to the Prosit webserver and converted to a spectrum library in generic text format (Gessulat 2019). The resulting library (20,374 protein isoforms, 28,307 protein groups, and 1,626,266 precursors) was used in DIA-NN 1.7.16 (Demichev 2020) to create a library directly from the acquired sample data using the MBR function. The applied settings were: output will be filtered at 0.01 FDR, N-terminal methionine excision enabled, maximum number of missed cleavages set to 1 min peptide length set to 7, max peptide length set to 30, min precursor *m/z* set to 400, max precursor *m/z* set to 1000, cysteine carbamidomethylation enabled as a fixed modification, and double pass search enabled.

### 2.6. Image Analysis and Statistical Analysis

For the mouse tissue section images and the HLE cell culture images, analysis was carried out with biological triplicates, each one with 3 images per test condition. Each image used for analysis comprised at least 500 cells. We noted that the images of the cells that were not irradiated or were treated with T4N5 contained more cells. Image analysis in FIJI was performed using DAPI to establish a mask of all the nuclei and to quantify the CPD fluorescence intensity within each nuclei using FIJI’s “mean grey value” measurements. The statistical comparison of the image data was performed via one-way ANOVA, with a *p*-value of <0.05 considered to be statistically significant. The representative images used in Figure 1, Figure 2 and Figure 3 are closeups of the original images that were used for signal quantification.

For the proteomic data, MS analysis of the HLE cell lysate digests was carried out with biological triplicate. The proteins with a *p*-value of <0.05 were considered to be statistically significant.

## 3. Results

### 3.1. T4N5 Repairs CPDs in Human Limbal Epithelial Cells

Firstly, the effect of UVB on the incidence of CPD lesions was established in human corneal epithelial cells. We observed that UVB irradiation did cause a significant increase in CPD intensity (Figure 1D) compared to the non-irradiated control (Figure 1A). We established in the same cells that the T4N5 repair enzyme was able to repair the CPD lesions and that the BioPorter vehicle had no repairing effect when used alone (Figure 1B,E). The T4N5 treatment of the irradiated cells reversed the CPD intensity to the levels of the non-irradiated controls (Figure 1C,F). The quantification of the CPDs via image analysis confirms that the CPD amounts are significantly reduced with the T4N5 treatment (Figure 1G).

**Figure 1 biology-12-00265-f001:**
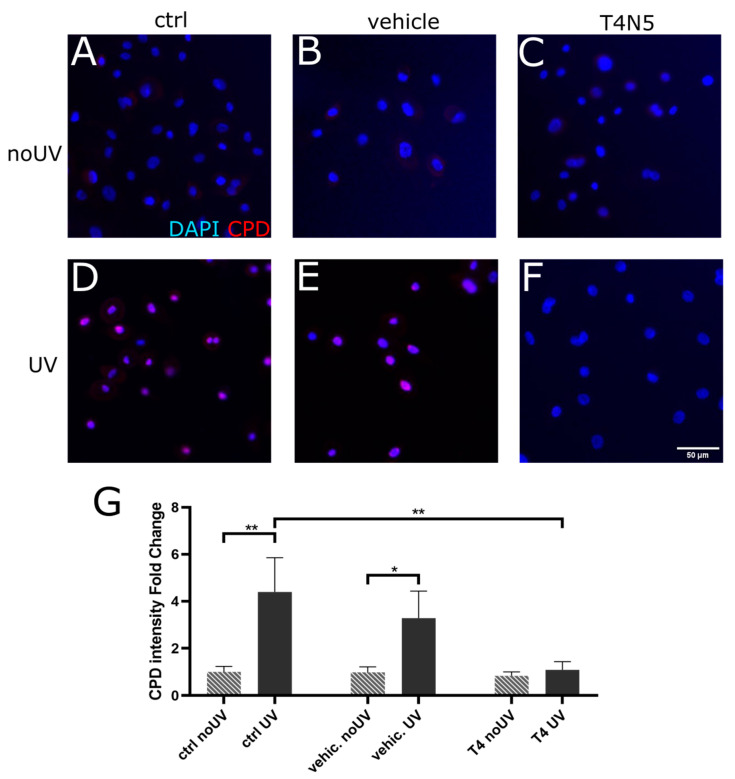
Mean fluorescence intensity of CPDs in human limbal epithelial cells; comparing untreated controls (**A**,**D**) (ctrl), vehicle controls (**B**,**E**) (vehic.), and T4 endonuclease V treatment (**C**,**F**) (T4). Cells were either irradiated with UVB (**D**–**F**) (UV) or not irradiated (**A**–**C**) (no UV). The mean fluorescence intensity of CPDs (**G**) was measured 48 h after 0.03 J/cm^2^ UVB; n = 3. CPD immunofluorescence was highest in irradiated cell that did not have T4 endonuclease V (*p* < 0.0001). * indicates *p* < 0.05, ** indicates *p* < 0.002.

### 3.2. T4N5 Repair Time Course in Human Limbal Epithelial Cells

To evaluate the rate of CPD repair by intrinsic mechanisms only as well as by treatment with T4N5, a time course of CPD intensity following UV damage was carried out over 96 h. The images and associated measurements showed that UV irradiation without the subsequent T4N5 treatment caused a significant increase in the CPD intensity, which was still observable up to 24 h post-irradiation (see Figure 2B–D). A significant reduction in the CPD signal was observed after 48 h (Figure 2E). The CPD signal was reduced to a level similar to the non-irradiated control after 72 h of incubation in the dark at 37 °C (see Figure 2A,F,H). In contrast, the T4N5 treatment directly after irradiation seems to lead to no significant increase in CPD intensity at the 12 h timepoint (see Figure 3C), suggesting that the enzyme had an immediate repairing effect as no CPDs could be detected at any timepoint beyond 12 h (Figure 3C–G). The only evidence of CPDs occurring was immediately after irradiation (Figure 1B). The CPD intensity returned to the non-irradiated levels within 12 h (Figure 3A,C,H).

**Figure 2 biology-12-00265-f002:**
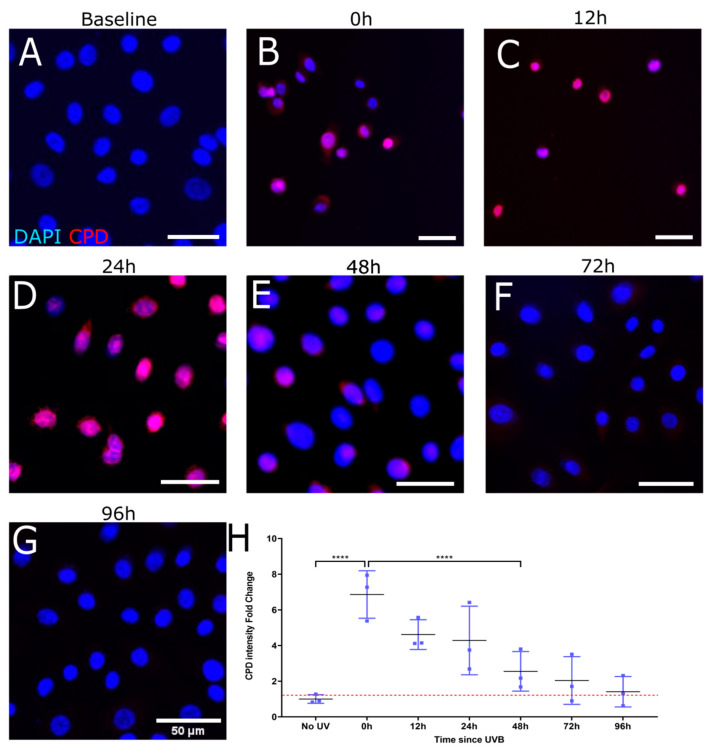
Representative immunofluorescence images illustrating recovery of induced CPD in HLE following UVB exposure. Merged images show nuclei stained blue (DAPI) and CPDs stained red. Mean fluorescence measurements were tracked over the course of 96 h after 0.03 J/cm2 UVB (**B**–**G**). A non-irradiated control was used for comparison (**A**) and to set a baseline when quantifying fold change in CPD immunofluorescence (**H**). CPD immunofluorescence was highest at 24 h (**** for *p* < 0.001) (**D**). Each data point was analyzed from 3 donors, each giving 3 images of over 500 cells per image.

**Figure 3 biology-12-00265-f003:**
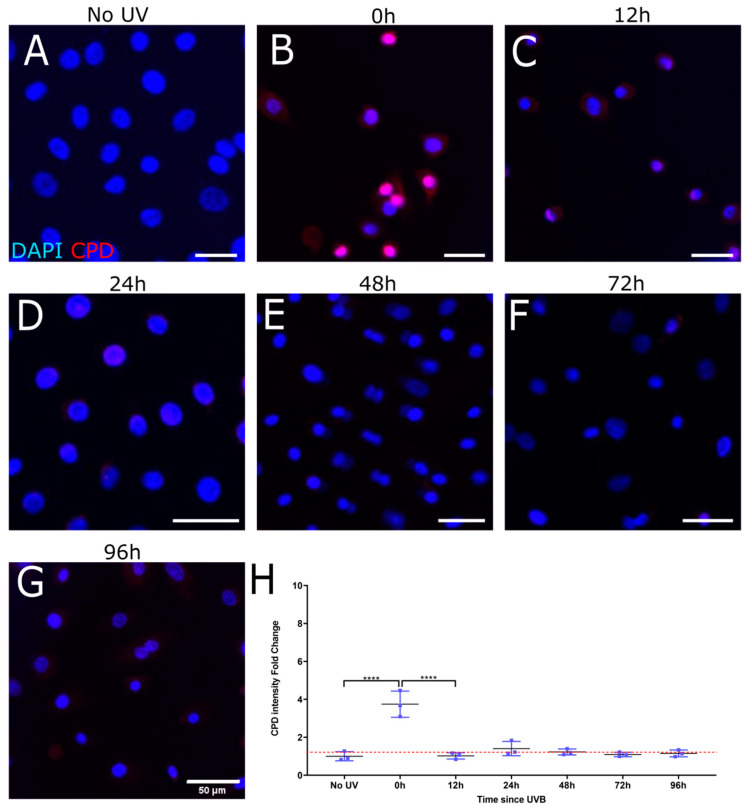
Representative immunofluorescence images illustrating recovery of induced CPD in HLE following UVB exposure and T4 endonuclease V treatment. Merged images show nuclei stained blue (DAPI) and CPDs stained red. Mean fluorescence measurements were tracked over the course of 96 h after 0.03 J/cm^2^ UVB (**B**–**G**). A non-irradiated control was used for comparison (**A**) and to set a baseline when quantifying fold change in CPD immunofluorescence (**H**). CPD immunofluorescence was highest at 0 h (**** for *p* < 0.001) (**B**). Each data point was analyzed from 3 donors, each giving 3 images of over 500 cells per image.

### 3.3. Proteomic Analysis of T4 Endonuclease Treatment

Equal quantities of whole cell lysates from HLE cultures were harvested following irradiation and treatment with T4 endonuclease V or a control. A total of 4754 proteins were quantified in all three biological donors (Figure 4A). A total of 1176 proteins were found to be significantly different when comparing the irradiated control to the irradiated, treated cells. The proteins were further subdivided into function-related groups, with two groups of particular interest: inflammation-related and angiogenesis-related. There were 21 inflammation-related proteins (Figure 4B) and 27 angiogenesis-related proteins (see Figure 4C), with 3 proteins appearing in both groups.

The highest increase in protein quantity due to the treatment was detected in PDCL3 (Phosducin-like 3, also known as PhLP2A), a protein present in several tissues performing several roles. One of these roles is the regulation of angiogenic factor VEGF via its receptor VEGFR-2 [55]. The greatest decrease in protein quantity was seen in erbB2 (erb-b2 receptor tyrosine kinase 2). The bone morphogenetic protein receptor type-2 (BMPR2) was also decreased following treatment with T4N5.

The inflammatory proteins were also generally more present following the treatment with T4N5. Bromodomain-containing protein 4 (Brd4) in particular was found in greater quantities following the treatment. Interleukin 1 (IL1A) and associated proteins such as interleukin 1 receptor antagonist (IL1RA) and interleukin 1 receptor accessory protein (IL1RAP) were also affected by T4N5. Along with IL1A, two more proteins that fit in the anigogenic and inflammatory groups were affected. These two proteins were: Aminoacyl-tRNA-synthetase-interacting multifunctional protein 1 (AIMP1) and IL18.

### 3.4. CPD Recovery in Sections of Whole Mouse Cornea

In order to verify that CPDPL was present in the cornea of the CPDPL tg/wt mice, staining of K14 was performed on the sections. This is because there are no readily available CPDPL antibodies, and it is also to verify that the protein produced by the K14 gene was indeed present in the corneal epithelium. The K14 staining gave a bright signal at the limbus and a clear signal in the basal corneal epithelium (Figure 5). This suggests that K14 is indeed being expressed and that the CPDPL gene placed downstream of the K14 gene should also be expressed. Further assessment of the capacity of CPDPL to repair the CPD lesions in mice was first performed using extracted and irradiated whole eye globes. The eyes were then either photorecovered or kept in the dark before fixing and sectioning. Without the photoreactivation step, which is essential for the activation of the CPDPL enzyme, only a non-significant difference could be detected between the wild-type controls and the CPDPL-producing mutants. When a 30 min photoreactivation step was included immediately after irradiation, a significant decrease in CPD intensity could be observed (Figure 6A–D). Notably, no significant difference was observed between the mutant CPDPL tg/wt mice that were photoreactivated and a mutant that was not photoreactivated. This may be due to how sensitive the enzyme is and capable of drawing the energy required for repair from photons in the smallest amount of light. Completely obscuring the control eyes proved challenging (Figure 6). There was a small but non-significant increase in CPD intensity in the photoreactivated CPDPL wt/wt mice; it is possible that 30 min under bright white light after a round of UVB irradiation marginally increases the CPD incidence (Figure 6E).

### 3.5. CPD Recovery in the Limbus of Whole Mouse Cornea

The whole mouse cornea, flattened and imaged (Figure 7A and Figure 8A), show that the p63a+ cells, the limbal stem cells (Figure 7D and Figure 8D), develop CPDs after UVB irradiation (Figure 7C and Figure 8C). CPDs also occur in epithelial cells outside of the limbus (Figure 7B,C). Photoreactivation removed CPDs from the cornea in the p63a-cells. However, the p63a-expressing cells featured a persistent CPD signal (Figure 8C highlighted by arrows). The photoreactivation results in the stem cell-housing limbal, as defined by the p63a staining, have showing incomplete CPD recovery in the limbus (Figure 8C). Notably, the addition of T4N5 appears to have reduced the CPD signal but did not lead to a full recovery of the limbus (Figure 9B,C). The counting of the CPD-positive nuclei allows for some quantification of the CPD clearance that each enzyme offers in the center and limbus of mouse cornea (Figure 10). It is apparent from the quantification that the CPDPL photoreactivation had the most positive effect on central CPD repair, with clear and significant improvement (Figure 9A and Figure 10, red). The use of T4N5 on the cornea did not have a significant effect on the T4N5 clearance in tissue. Interestingly, the CPDPL photoreactivation did not have a significant effect on the central CPD when paired with T4N5. Overall, the limbal CPD amount was not significantly affected by any of the treatments (Figure 10, blue).

## 4. Discussion

CPD repair was achieved in both the cell cultures and the ex vivo tissue. Further analysis in vitro revealed increased inflammation and angiogenesis. Further work in the ex vivo tissue revealed that the total clearance of CPDs in the limbus was not achieved.

First, we aimed to identify whether the DNA repair enzyme T4N5 would facilitate the DNA damage repair in corneal epithelial cells. The repair of the photolesions in vitro, as assessed by CPD quantification, shows that cells can repair CPDs naturally within 72 h of the irradiation event. The addition of T4N5 effectively removes CPDs, resulting in a similar anti-CPD antibody staining level as that of the baseline non-irradiated cells. The kinetics of the endogenous repair of CPDs is slow and the 72 h during which the cells contain a significant number of photolesions might be detrimental. Indeed, during that time a CPD lesion might lead to a mutation when, for instance, the cell cycle is not completely arrested amid the damage. This is particularly relevant in the cases where the cells need to divide, such as in wound healing. Modern paradigms of tissue healing will often aim to support endemic wound healing responses as well as introduce novel repair solutions. While the cells seem capable of repairing CPDs without the use of exogenous enzymes, the addition of T4N5 greatly improves the CPD repair rate. T4N5 provides a faster clearing of CPDs than what the cell is normally capable of, as well as preventing mutations during the critical phases of rapid division.

Then, we aimed to identify any effect the T4N5 treatment may have had on the cell behavior, with a particular interest in any inflammatory and angiogenic effects. The proteomic analysis of the UV-irradiated HLE cells showed that the T4N5 treatment increased the amounts of angiogenic and inflammatory proteins in HLE. It is a possibility that, even though the treated cells show increased quantities of angiogenic and inflammatory proteins, it is a more desirable outcome. The cells that would normally die after irradiation may have survived instead, due to the T4N5 treatment. The analysis of the proteomic data revealed that 165 apoptosis-related proteins were significantly different following the treatment and of those proteins, 6 were downregulated following treatment. One of these proteins also shows up in the angiogenesis-related group of interest: BMPR2. BMPR2, along with BMP-family proteins and their downstream dependents, such as the Smad-family proteins, are certainly affected by this. BMPs and the Smad-family proteins have varied functions, making the exact effects of the changes observed here uncertain [56,57]. Bone morphogenic protein (BMP) binds to bone morphogenetic protein receptor type-1 (BMPR1) in the absence of BMPR2. When both BMPR1 and BMPR2 are present, their binding affinity increases greatly [58]. The reduction in the BMPR2 quantity suggests that, while the BMPR1 amount did not change significantly, the BMPR1 activity may be increased following T4N5 treatment. The presence of BMPR1 and BMPR2 mRNA in the corneal epithelium has been recorded [59]. It is important to note that cell culture reportedly alters the BMP signaling of corneal cells, with cultured cells differentiating more often into BMP-driven neural lineages [60]. The signaling pathway downstream from BMP, such as the Smad family proteins, may be affected the most by this change in the BMPR2 presence. Moreover, the observed elevation of PDCL3 possibly leads to the ligand-stimulated phosphorylation of VEGFR-2 and the activation of VEGFR-2 substrate, PLCγ1. Protein erbB2 is present in several tissues and, in its role as receptor tyrosine kinase, is involved in epithelial growth factor receptor signaling [61]. In the corneal epithelium, erbB2 is linked to post-injury cell migration [62]. The great decrease in erbB2 noted after the T4N5 treatment suggests that cell migration is affected, although the effect is unclear.

The apparent increase in angiogenic and inflammatory activity following T4N5 treatment may be undesirable for corneal treatments. However, it should be noted that previous work in similar conditions observed that the UVB irradiation of HLE downregulated the pro-angiogenic proteins. The increase in angiogenic factors we observe may simply be a return to normal.

An increase in inflammatory factors was also observed. The greatest increase was noted in Brd4. Brd4 is a histone-binding major epigenetic regulator that maintains the chromatin structure of descendent cells [63,64,65]. One of the more important functions of Brd4 is cell identity determination and stem cell maintenance in several tissues [66,67,68]. In the cornea, Brd4 has been reported as a potential epithelial stem cell marker due to its presence in the less differentiated cells of the basal and limbal epithelium [69]. Brd4 has also been reported as a regulator of fibrotic scarring in corneal scarring, with particular interest towards its suppression as a clinical target [70]. It possible that T4N5 treatment preserves stem cell identity in the HLE while also driving a greater fibrotic response.

IL1RAP was reduced after treatment; IL1RAP acts as an immune mediator to regulate pro-inflammatory and mitogenic signaling pathways [71]. In the cornea, upregulation of IL1RAP is reported as having some correlation to dry eye disease, with the subsequent reduction in IL1RAP accompanying disease treatment [72]. IL1RA is an anti-inflammatory cytokine and is also reported as having conditionally anti-angiogenic capabilities in the cornea [73,74,75]. Elevated IL1A in response to UV irradiation is a well-documented response, particularly in the cornea [76,77]. The inflammatory cytokine is also a marker of the corneal angiogenic response to injury [78]. IL1A is involved in several of the steps associated with angiogenesis, such as acute inflammation, inflammatory cell activation, and chemotactic recruitment, and adhesion molecule upregulation [79,80]. The two other significantly affected proteins that have both angiogenic and inflammatory proteins were IL18 and AIMP1.

IL18 is an inflammatory cytokine known to be upregulated in the corneal epithelium following UVB irradiation [81,82]. The increase seen after the T4N5 treatment suggests that the inflammation reaction is greater. Commonly associated inflammatory cytokines such as IL1B and IL6 were not significantly increased or decreased. This partial increase in inflammation after T4N5 treatment may be due to more cells surviving UVB irradiation, allowing cells that would normally die to instead secrete inflammatory cytokines.

AIMP1 can form complexes with several other proteins, one of which is the secreted cytokine p43. When in a low p43 concentration, the AIMP1-p43 complex will drive MMP9 activity to facilitate angiogenesis. When in high p43 concentrations, the same complex with induce endothelial cell apoptosis via Jun N-terminal kinase activation [83,84]. This bimodal, p43-dependent activity of AIMP1 makes any conclusion about the effect of high AIMP1 on vascularization post-T4N5 treatment impossible, especially since there was no significant difference in the p43 quantity. AIMP1 is also an activator of monocyte and macrophages.

Moving to the ex vivo approaches, we aimed to identify whether CPD photolyase, T4N5, or a combination of both would improve CPD clearance. The CPD photolyase introduction into mice proved successful in repairing CPD, with significant improvements in CPD reduction, as estimated by the image quantification. Similar to T4N5, CPD photolyase offers a reparative solution to cells, although here it was demonstrated ex vivo. The differences of note between T4N5 and CPDPL are that CPDPL requires a photoreactivation step and only targets CPD bonds. The documented higher specificity of the photolyase may make it more desirable, although T4N5 is documented extensively as having no known side effects [42,43]. The photoreactivation step may be of some concern in cases where the eye is not exposed to light, but we note that an exposure of 30 min to simple fluorescent light, such as that found in offices and homes, is enough to trigger significant repairs.

The localization of CPD lesions proved to be particularly interesting in the ex vivo irradiated eyes. While the central epithelial cells did acquire CPD lesions that could be removed following the photoreactivation of CPDPL, the limbal epithelial cells had CPDs that could not be repaired. The immunofluorescent imaging of K14 shows that the K14Cre driver is indeed active in the entire corneal epithelium, especially in the limbus. We can reasonably expect CPDPL to be present in the same locations. Future work may aim to incorporate techniques such as RNA FISH to confirm this. We concluded that we have achieved our goal of improved CPD clearance, with the faster removal of CPDs in cells and fewer CPD lesions in the ex vivo tissue. The implications of the increased inflammation and the angiogenesis, as well as the incomplete limbal clearance of CPDs, remain to be studied, including in human tissue.

## 5. Conclusions

Our study allows the following conclusions to be drawn: (a) T4N5 can repair CPDs in cultured cells. (b) T4N5 enhances the angiogenic and inflammatory activity of HLE in vitro. (c) CPDPL photorecovery can repair CPDs in the CPDPL mutant mice. (d) CPDPL photorecovery does not repair CPDs in the limbal p63a-positive cells of the CPDPL mutant mice.

One of the greatest risk factors for the development of ocular surface tumors is UV exposure. The damage that UVB causes in the corneal epithelium can be mitigated with the use of exogenous repair enzymes such as CPDPL and T4N5. T4N5 in particular can assist existing mechanisms, allowing repairs that would normally take several days to instead be complete within hours. There is clinical potential in the application of these enzymes on vulnerable corneas, particularly in post-surgery environments where UVB damage is a possibility.

## Figures and Tables

**Figure 4 biology-12-00265-f004:**
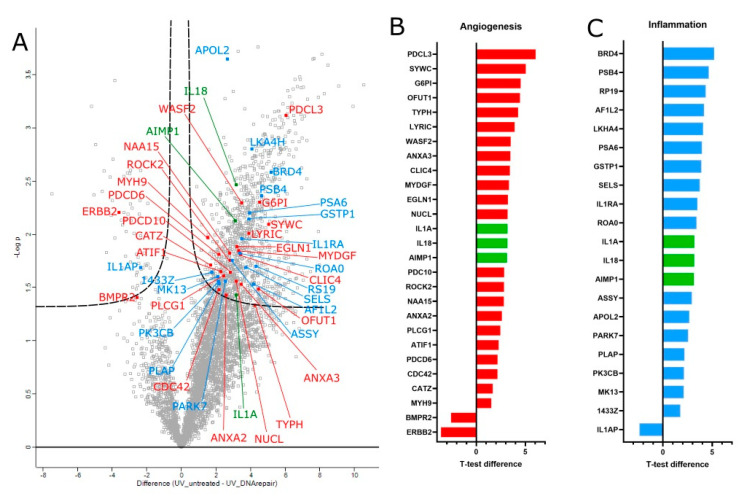
(**A**) Volcano plot analysis for differentially expressed proteins in human limbal epithelial cells across three biological replicates, comparing UV irradiated with and without T4 endonuclease V treatment. The proteins are labelled with their gene names. The difference in the means of the expressed proteins on the *x*-axis vs. the statistical significance (−log *p*-value) on the *y*-axis is shown. Out of the 1176 statistically significant proteins, 20 inflammation-related proteins (blue) and 26 angiogenesis proteins (red) were highlighted. Three proteins were found to be involved in both functions (green). The *T*-test difference values for the two groups of interest are represented in a bar chart for angiogenesis-related proteins (**B**) and inflammatory-related proteins (**C**).

**Figure 5 biology-12-00265-f005:**
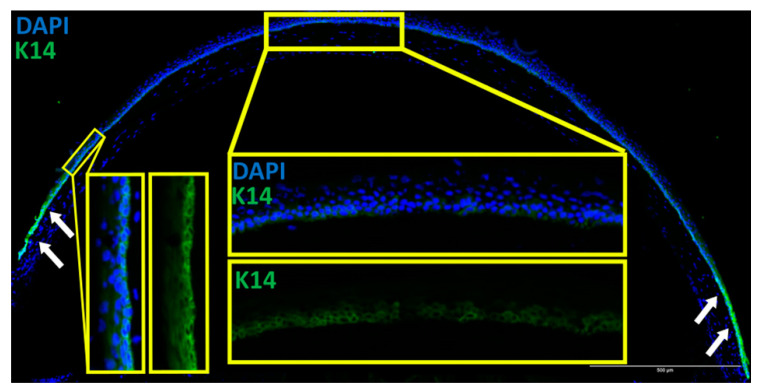
K14 stain of a K14-CPD-PL mouse cornea section, showing clear K14 expression in the epithelial basal layer in the central cornea and expression across the full thickness of the epithelium in the limbus. The bright signal shown at the limbus (arrows) demonstrates high presence of K14.

**Figure 6 biology-12-00265-f006:**
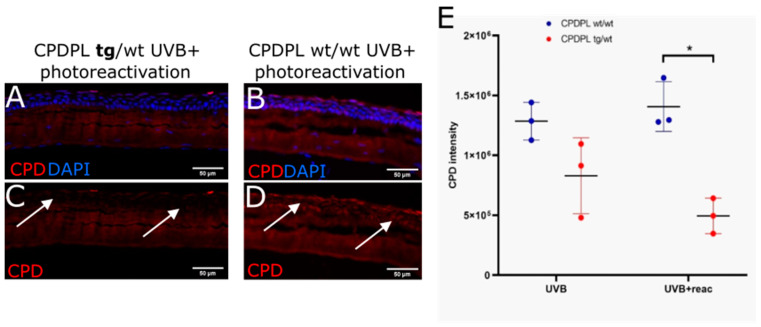
Representative immunofluorescence images illustrating recovery of CPDPL mutant mice following UVB exposure and photoreactivation. Merged images (**A**,**B**) show nuclei stained blue (DAPI) and CPDs stained red (positive nuclei indicated by arrows in (**D**) and absence of signal indicated by arrows in (**C**). (**E**) Mean fluorescence measurements were taken after 1 J/cm^2^ UVB and 30 min photoreactivation. CPD immunofluorescence was highest in irradiated as well as photoreactivated wt/wt (* is *p* < 0.05). Each data point was analyzed from 3 technical repeats.

**Figure 7 biology-12-00265-f007:**
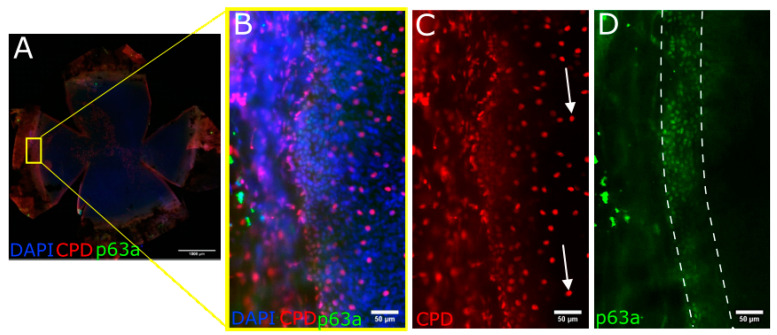
Images illustrating CPD induction in control mice following UVB exposure. CPDs stained with Alexa555 ((**A**–**C**) shown here in red) and stem cell marker p63a stained with Alexa488 ((**A**,**B**,**D**) shown here in green). Cells positive for CPDs outside the limbus are indicated by arrows in (**C**). Images were taken after 1 J/cm^2^ UVB. CPD immunofluorescence was particularly concentrated in limbal regions, although not brighter.

**Figure 8 biology-12-00265-f008:**
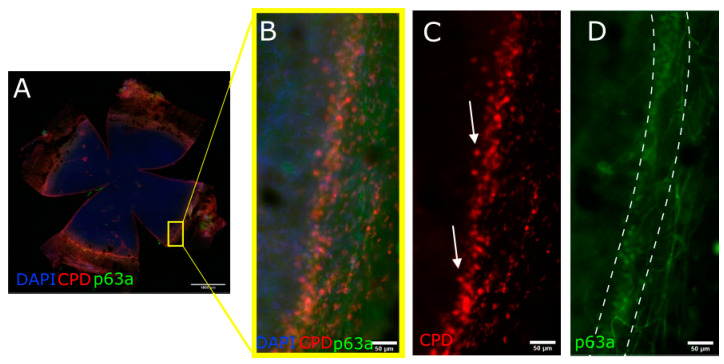
Images illustrating CPD recovery in CPDPL mutant mice following UVB exposure and 30 min photorecovery. CPDs stained with Alexa555 ((**A**–**C**) shown here in red) and stem cell marker p63a stained with Alexa488 ((**A**,**B**,**D**) shown here in green). Cells positive for CPDs inside the limbus are indicated by arrows in (**C**). Images were taken after 1 J/cm^2^ UVB 30 min photorecovery. CPD immunofluorescence was particularly concentrated in limbal region and generally absent in central cornea.

**Figure 9 biology-12-00265-f009:**
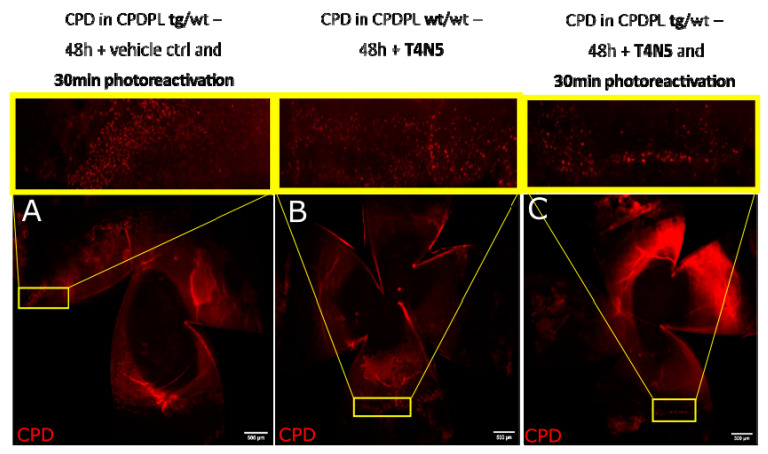
Images illustrating recovery of induced CPDs in control mice and CPDPL mutant mice following UVB exposure and either photoreactivation and 48 h with vehicle control (**A**) and T4N5 (**B**), or T4N5 and 30 min photoreactivation (**C**). CPDs stained red. Images were taken after 1 J/cm^2^ UVB and 48 h of incubation. CPD immunofluorescence was highest in irradiated tissue treated with vehicle control.

**Figure 10 biology-12-00265-f010:**
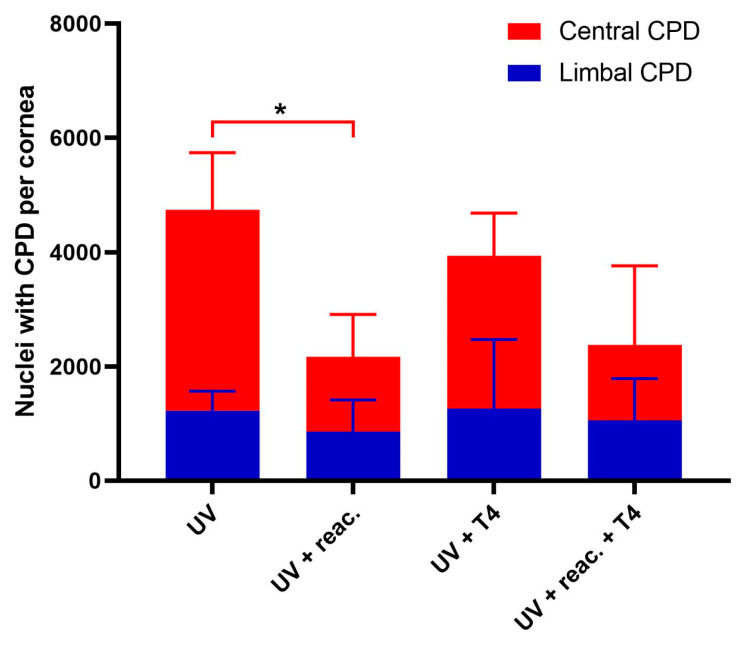
Bar chart of the number of CPD-positive nuclei in irradiated CPDPL tg/wt mouse corneas. Each column represents the total number of CPD-positive nuclei in the cornea, the red portion represents the nuclei in the center of the cornea, and the blue portion represents the nuclei in the limbus. There was no significant change in limbal CPDs noted after treatment with T4N5 or activation of CPDPL. Central CPD change was only noted after photoreactivation (* is *p* < 0.05).

## Data Availability

Publicly available proteomics datasets were analyzed in this study. This data can be found here: https://www.ebi.ac.uk/pride/, (accessed on 11 January 2023). Website cited in [85].

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
