# Peer review of "Short-Term UVB Irradiation Leads to Persistent DNA Damage in Limbal Epithelial Stem Cells, Partially Reversed by DNA Repairing Enzymes"

_biology, 2023, doi:10.3390/biology12020265_

Round 1

Reviewer 1 Report

In this manuscript, the authors investigated the repair efficiency of exogenous T4N5 and/or CPD photolyase on CPD lesions in cultured cells and mouse models. The results showed that T4N5 could repair CPD lesions in cultured HLE cells and CPDPL could repair CPD lesions in mouse models. This work demonstrates the potential of CPDPL and T4N5 for the treatment of CPD lesion in limbus and provides novel ideas to treat cornea photolesions. Below are some suggestions/comments.

Issues:

Image analysis and Statistical analysis: How was the CPD fluorescence intensity measured? I would suggest the authors average the CPD intensity among cells to get more accurate quantifications results.

Figure 2 & 3: authors stated that “each giving 3 images of over 500 cells per image”, while according to the displayed figures, the number of the cells is less than 500. Were different figures used for statistical analysis?

Figure 4A: For the volcano plot, I would recommend using a line of dashes on the graph to indicate the cut-off of p-value, which is -Log p. In addition to the p-value, a cut-off value such as 1.5-fold or 2-fold should be set for the protein fold change between the two conditions. In this way, only when the fold change of the protein was larger than 1.5 or 2 with a p-value < 0.05, the protein would be considered significantly changed under T4N5 treatment. Proteins can be classified into significant up-regulated and down-regulated groups and then analyzed with Gene Oncology and KEGG pathway, which can facilitate the study of the function of T4N5 in CPD repair.

Figure 6: How were the three technical replicates conducted? Three repeats with the same sample at different regions or the same regions?

Figure 9: According to the figures, the CPD intensity is almost unchanged under T4N5 treatment, which means T4N5 only did not significantly repair the CPD in limbus.  Does it mean it is CPDPL that is mainly responsible for the repair of CPD in limbus? Why didn’t the authors perform the experiments with CPDPL tg/wt mice with 30 min photoreactivation to see the repair efficiency of CPDPL only?

Minor issues:

Page 3, line 39: Change “These were leftover tissue from surgery.” to “These were leftover tissues from surgery.”

Page 4, line 1: Change “…80uL…” to “…80 µL…”

Page 4, line 8: Change “2*104” to “2*104”; “cm2” to “cm2

Page 5, line 23: Change “UltiMate 3000 nLC” to “UltiMate 3000 HPLC” or “UltiMate 3000 UHPLC”

Page 5, line 28: Change “6&” to “6%”

Page 12, line 8: Change (fig. 8C)..” to (fig. 8C).”

Figure 1, Legend was not seen.

Figure 3, Legend: Change “p > 0.001” to “p < 0.001”

Figure 6, Legend: Change “p >0.05” to “p < 0.05”. Please label the left and right panels with A and B to distinguish them.

Author Response

Dear Reviewer,

Thank you for considering our manuscript  ‘Short-term UVB irradiation leads to persistent DNA damage in limbal epithelial stem cells, partially reversed by DNA repairing enzymesfor publication in Biology.

We have read carefully the comments you have made and we are providing a point to point response below. We have addressed all concerns and incorporated your suggestions. We hope that these modifications have strengthened our manuscript and enhanced its clarity.

Best wishes

Thomas Volatier, PhD

Cornea Lab

Experimental Ophthalmology

Dept. of Ophthalmology

University of Cologne

tel. +49-(0)221-478-97793

[email protected]

In this manuscript, the authors investigated the repair efficiency of exogenous T4N5 and/or CPD photolyase on CPD lesions in cultured cells and mouse models. The results showed that T4N5 could repair CPD lesions in cultured HLE cells and CPDPL could repair CPD lesions in mouse models. This work demonstrates the potential of CPDPL and T4N5 for the treatment of CPD lesion in limbus and provides novel ideas to treat cornea photolesions. Below are some suggestions/comments.

Issues:

Image analysis and Statistical analysis: How was the CPD fluorescence intensity measured? I would suggest the authors average the CPD intensity among cells to get more accurate quantifications results.

The CPD fluorescence intensity was measured by first detecting each nucleus using DAPI. Then, within each nucleus, a CPD measurement was taken. This gives a list of several measurements, each from a single cell. We then averaged the intensity readings for each donor. Then, the three donor readings were averaged to reach the final intensity value.

In methodology 2.6, we added the sentence “Each image used for analysis comprised at least 500 cells. We noted that images of cells that were not irradiated or were treated with T4N5 contained more cells.”

In methodology 2.6, we clarified “Image analysis in FIJI was performed using DAPI to establish a mask of all nuclei and quantify CPD fluorescence intensity within each nuclei using FIJI’s “mean grey value” measurements”

In Methodology 2.4, specified, “All images used for intensity measurements were taken with the same settings on the same day.”

Figure 2 & 3: authors stated that “each giving 3 images of over 500 cells per image”, while according to the displayed figures, the number of the cells is less than 500. Were different figures used for statistical analysis?

The images used for statistical analysis are much larger to included greater numbers of cells. The images used in figures 1, 2, and 3 are representative closeups of the cells to show CPD signal in the nuclei to the reader.

In methodology 2.6, we clarified “Representative image used in Figures 1, 2, and 3 are closeups of the original images that were used for signal quantification.”

Figure 4A: For the volcano plot, I would recommend using a line of dashes on the graph to indicate the cut-off of p-value, which is -Log p. In addition to the p-value, a cut-off value such as 1.5-fold or 2-fold should be set for the protein fold change between the two conditions. In this way, only when the fold change of the protein was larger than 1.5 or 2 with a p-value < 0.05, the protein would be considered significantly changed under T4N5 treatment. Proteins can be classified into significant up-regulated and down-regulated groups and then analyzed with Gene Oncology and KEGG pathway, which can facilitate the study of the function of T4N5 in CPD repair.

Thank you for your suggestion, we added a volcano plot statistical analysis in the form of 2 dashed lines separating the points on figure 4. All fold changes listed have a T-test difference of 1 or more. A difference of "1" means a fold change of 2, whereas a difference of -1 means a fold change of two t the left group of the test (or factor 0.5, depending on the argumentation). All proteins mentioned in Figure 4 have a fold change greater than 2.

Figure 6: How were the three technical replicates conducted? Three repeats with the same sample at different regions or the same regions?

Apologies for not being more clear about this. Yes, the three technical repeats are three different locations within the same cornea. CPD repair effectiveness varies in the cornea so both limbus and central cornea were imaged to quantify CPD intensity. Three biological repeats were used.

In methodology 2.4, we specified, “For mouse cornea sections, three images where taken for each biological repeat. These technical repeats imaged the limbus as well as the center of the cornea.”

Figure 9: According to the figures, the CPD intensity is almost unchanged under T4N5 treatment, which means T4N5 only did not significantly repair the CPD in limbus.  Does it mean it is CPDPL that is mainly responsible for the repair of CPD in limbus? Why didn’t the authors perform the experiments with CPDPL tg/wt mice with 30 min photoreactivation to see the repair efficiency of CPDPL only?

We apologise for the confusion. Figure 9A originally was labelled with “as being a cornea from a CPDPL wt/wt mouse”, but is in fact from a mutant CPDPL tg/wt mouse that underwent photoreactivation. The orginal Figure 9A was mislabeled as a non irradiated wildtype control. This has been rectified and Figure 9A has been relabeled as “CPD in CPDPL tg/wt – 48h + vehicle ctrl and 30min photoreactivation”

Experiments were performed on CPDPL tg/wt mice with 30 min photoreactivation. The results are shown in the newly added Figure 10 where a significant decrease in central CPD can be seen.

Minor issues:

Page 3, line 39: Change “These were leftover tissue from surgery.” to “These were leftover tissues from surgery.”

Changed “These were leftover tissue from surgery.” to “These were leftover tissues from surgery.”

Page 4, line 1: Change “…80uL…” to “…80 µL…”

Changed “80uL” to “80 µl”

Page 4, line 8: Change “2*104” to “2*104”; “cm2” to “cm2

Changed “density of 2*104 cells per cm2” to “density of 2*104 cells per cm2

Changed every instance of “cm2” to “cm2

Page 5, line 23: Change “UltiMate 3000 nLC” to “UltiMate 3000 HPLC” or “UltiMate 3000 UHPLC”

Changed “UltiMate 3000 nLC” to “UltiMate 3000 HPLC”

Page 5, line 28: Change “6&” to “6%”

Changed “6&” to “6%”

Page 12, line 8: Change (fig. 8C)..” to (fig. 8C).”

Removed the extra period after “(fig. 8C).”

Figure 1, Legend was not seen.

Apologies for this. Figure Legend should be visible now

Figure 3, Legend: Change “p > 0.001” to “p < 0.001”

Changed “p > 0.001” to “p < 0.001”

Figure 6, Legend: Change “p >0.05” to “p < 0.05”. Please label the left and right panels with A and B to distinguish them.

Changed “p >0.05” to “p < 0.05”. Each image in figure 6 is labelled A to D. The graph in figure 6 is labelled E.

Reviewer 2 Report

In the present work, the authors detected the expression of CPD, a marker of DNA photolesion, in the corneal limbal epithelial stem cells after UVB damage. In addition, they also evaluated the efficacy of CPD photolyase and T4N5 on repairing CPD lesion from the limbus after UVB exposure. The study was well designed and analyzed. However, there are several improvements should be made prior to publication

1. In 2.2 of materials and methods section, how do you determine the power of UVB? How you determine the concentration of T4N5 in vitro and ex vivo?

2. In Figure 1G, Figure 2H and Figure 3H, the fluorescence intensity of CPD can be presented as fold change. In addition, how you calculate the fluorescence intensity? How to select the number of cells?

3. The scale of cells is obviously different in Figure 2 and Figure 3. For example, Fig. 2A and 2C; Fig. 3A and 3C, etc.

4. In 3.3 of ‘Result’ section, Is there any western blot verification after proteomic analysis? In addition, it needs to be further clarified which differential proteins are up-regulated or down-regulated compared with control after T4N5 treatment? In addition, how is the action time of T4N5 determined?

5. Figure 6E showed that the CPD fluorescence intensity at the limbus of the CPDPL tg/tw mice decreased significantly after photoactivation, but the result (lines 354-356) said there was no significant difference, please explain the reason.

6. It is suggested that the fluorescence intensity in Figure 7-9 should be numerical and counted. Among them, the CPD intensity of Figure 7 and Figure 8 need to be statistically compared.

7. The CPD fluorescence intensity of three figures in Figure 9 should be compared.

8. In ‘Discussion’ section, although the possible role of differential inflammatory factors and angiogenesis related factors in T4N5 treatment of limbal epithelial cells was explained in the discussion, there was no validation in vivo or in vitro. Please further prove the possible mechanism.

Author Response

Dear Reviewer,

Thank you for considering our manuscript  ‘Short-term UVB irradiation leads to persistent DNA damage in limbal epithelial stem cells, partially reversed by DNA repairing enzymesfor publication in Biology.

We have read carefully the comments you have made and we are providing a point to point response below. We have addressed all concerns and incorporated your suggestions. We hope that these modifications have strengthened our manuscript and enhanced its clarity.

Best wishes

Thomas Volatier, PhD

Cornea Lab

Experimental Ophthalmology

Dept. of Ophthalmology

University of Cologne

tel. +49-(0)221-478-97793

[email protected]

In the present work, the authors detected the expression of CPD, a marker of DNA photolesion, in the corneal limbal epithelial stem cells after UVB damage. In addition, they also evaluated the efficacy of CPD photolyase and T4N5 on repairing CPD lesion from the limbus after UVB exposure. The study was well designed and analyzed. However, there are several improvements should be made prior to publication

  1. In 2.2 of materials and methods section, how do you determine the power of UVB? How you determine the concentration of T4N5 in vitroand ex vivo?

Apologies for not clarifying this. In section 2.2 of materials and methods, we added the sentence “UVB intensity was measured by the machine’s internal sensor.” Specified concentration of T4N5 stock in “T4 endonuclease stock at a concentration of 10.000 units/ml” as well as final concentration of cells in “360μl of Imaging solution (Gibco) was added to the BioPorter tube for  T4N5 concentration of 500 units/ml.”

  1. In Figure 1G, Figure 2H and Figure 3H, the fluorescence intensity of CPD can be presented as fold change. In addition, how you calculate the fluorescence intensity? How to select the number of cells?

To address this concern and make the section more clear we changed graphs in figure 1G, 2H, and 3H to present CPD fluorescence intensity as fold change.

We specified in Methodology 2.4 that “All images used for intensity measurements were taken with the same settings on the same day.” Furthermore we specified in methodology 2.6 that “To quantify CPD fluorescence intensity within each nucleus we used FIJI’s “mean grey value” measurements”.

We specified in methodology 2.6 that “Each image used for analysis comprised at least 500 cells. We noted that images of cells that were not irradiated or were treated with T4N5 contained more cells on average.”

  1. The scale of cells is obviously different in Figure 2 and Figure 3. For example, Fig. 2A and 2C; Fig. 3A and 3C, etc.

We have added scale bars to all images in Figure 2 and 3.

  1. In 3.3 of ‘Result’ section, Is there any western blot verification after proteomic analysis? In addition, it needs to be further clarified which differential proteins are up-regulated or down-regulated compared with control after T4N5 treatment? In addition, how is the action time of T4N5 determined?

The scope of the experiment was high volume analysis of multiple proteins. Focused analysis of specific proteins either via western or ELISA will be a future step within follow up manuscripts.

We added a sentence in methodology 2.2 stating “The action time of T4N5 was decided based on manufacturer instruction for BioPorter and pilot experiments with T4N5.”

  1. Figure 6E showed that the CPD fluorescence intensity at the limbus of the CPDPL tg/tw mice decreased significantly after photoactivation, but the result (lines 354-356) said there was no significant difference, please explain the reason.

Figure 6E doesn’t show a significant difference between CPDPL tg/wt mice with and without photoreactivation. The photoreactivation could not significantly reduce CPD in CPDPL tg/wt mice because the CPD was being repaired even without photoreactivation. The curative enzyme CPDPL was being activated without photoreactivation.

The significant difference seen was between the CPDPL wt/wt control mice and the CPDPL tg/wt mice. Once both were exposed to photoreactivation, significantly more clearance of CPD was observed in the CPDPL tg/wt mice compared to the CPDPL wt/wt control mice.

  1. It is suggested that the fluorescence intensity in Figure 7-9 should be numerical and counted. Among them, the CPD intensity of Figure 7 and Figure 8 need to be statistically compared.

Thank you for your comment. We performed an analysis to measure CPD clearance in situ.

Specifically, the total number of CPD positive nuclei was counted to compare central cornea to limbal cornea and give a general idea of how successful each treatment was. The results have been graphed in the newly added Figure 10. This allows for the statistical comparison of CPD incidence in mice with CPDPL activation and mice without CPDPL activation (as is shown in figures 7 and 8)

  1. The CPD fluorescence intensity of three figures in Figure 9 should be compared.

Thank you for your suggestion. The newly added Figure 10 shows total number of CPD positive nuclei in the corneas subjected to the different treatments shown in Figure 9. The figure shows statistical change in central CPD amount but no significant change in limbal CPD amount. This allows for quantitative comparison of CPDPL, T4N5, and hybrid treatments. Specialized equipment is under assembly to repeat the ex vivo experiments in vivo. This will be the subject of our next publication on the topic.

  1. In ‘Discussion’ section, although the possible role of differential inflammatory factors and angiogenesis related factors in T4N5 treatment of limbal epithelial cells was explained in the discussion, there was no validation in vivoor in vitro. Please further prove the possible mechanism.

Specialized equipment is under assembly to repeat the ex vivo experiments in vivo. This will be the subject of our next publication on the topic.

Reviewer 3 Report

In this manuscript, the authors examine the effects of non-mammalian DNA repair enzymes in the repair of UV-induced CPDs in human epithelial cells and mouse corneas. Though photolyase and T4 endonuclease V have been employed in the skin, less work has been done in the context of the eye. Thus, this work is relevant to that tissue and potentially important. The work is for the most part adequately described, though I do have some suggested areas for improvement to make the manuscript clearer to readers.

Major concerns/comments:

  1. The authors have many figure panels (for example Fig. 1A, B) that are not referred to in the text. For Figs. 1-3, 6, it might make more sense to put all of the images as "Fig. #A" rather than labeling each image as a separate sub-panel.
  2. The authors should include sub-panel labels (A, B, C, etc.) and relevant descriptions in the figure legends.
  3. In Fig. 1G, what does 0X and 2X refer to? This should be included in the figure legend.
  4. The figure legend for Fig. 1 appears incomplete (it ends in p>).
  5. I don't understand the authors' statement "Staining of K14 was performed on sections to verify that the CPDPL promoter was indeed present in the corneal epithelium.". How does immunostaining for the K14 protein indicate the K14 promoter DNA element was present? Is the K14 not normally expressed in corneal epithelium, or is it unique to transgenic mice? I assumed the CPDPL expression was unique to the transgenic mice and expressed in the corneal epithelium, not the promoter that drives CPDPL expression.
  6. On line 375, the authors state "However, p63a-expressing cells featured persistent CPD signal (fig. 7C highlighted 375 by arrows)". However, though the locations of these cells (indicated by arrows in Fig. 7C) are not indicated by arrows in Fig. 7D, they do not appear to stain positive for p63. Are these cells negative for p63? Perhaps the authors can provide arrows in both Figs. 7C and D to be more clear about which p63-expressing cells the authors are wishing to point out.
  7. In lines 491-493, the authors state that "the repair proteins would need to unwind the histones to access lesions, something unlikely to happen outside of cell division." First, the histones are not "unwound". Rather, they are displaced from the DNA to allow access of DNA repair proteins to UV photoproducts in DNA. Second, decades of research have shown that UV photoproducts can be removed from DNA in the absence of cell division, and thus the authors' statement that DNA repair proteins would not be able to access DNA outside of cell division is not correct.
  8. In figures 2, 3, and 6, the authors indicate the level of statistical significance as p>0.001 or 0.05. Do the authors mean to say less than (<)? If not, perhaps the authors should better clarify the comparisons that are being made here.
  9. In Figure 6, the authors should indicate what the arrows represent in the figure legend.
  10. In Figure 9, is it possible for the authors to provide some type of quantitation showing the purported differences in CPD levels?

Minor comments:

  1. Line 79: It is called nucleotide excision repair, not nuclear excision repair.
  2. Line 158: The abbreviation HLE should be defined the first time it is used.
  3. Line 158: the 4 should be superscript
  4. In several places in the manuscript (including line 158), the 2 in "cm2" should be superscript.
  5. Line 174: x g, not G
  6. Line 214: the word diluted is repeated
  7. Line 223: mto?
  8. Line 349: were, not where
  9. Line 376: containing is misspelled
  10. Line 378: there are two periods at the end of the sentence.
  11. Line 483: insert ref should be fixed
  12. The abbreviation CPDPL should be defined somewhere in the manuscript. Presumably it means CPD photolyase.
  13. In some cases, the authors capitalize the 'F' in "Fig. x", but in others they do not. The authors should be consistent, and preferably capitalize it.

Author Response

Dear Reviewer,

Thank you for considering our manuscript  ‘Short-term UVB irradiation leads to persistent DNA damage in limbal epithelial stem cells, partially reversed by DNA repairing enzymesfor publication in Biology. We have addressed all concerns and incorporated your suggestions. We hope that these modifications have strengthened our manuscript and enhanced its clarity.

We have read carefully the comments you have made and we are providing a point to point response below.

Best wishes

Thomas Volatier, PhD

Cornea Lab

Experimental Ophthalmology

Dept. of Ophthalmology

University of Cologne

tel. +49-(0)221-478-97793

[email protected]

In this manuscript, the authors examine the effects of non-mammalian DNA repair enzymes in the repair of UV-induced CPDs in human epithelial cells and mouse corneas. Though photolyase and T4 endonuclease V have been employed in the skin, less work has been done in the context of the eye. Thus, this work is relevant to that tissue and potentially important. The work is for the most part adequately described, though I do have some suggested areas for improvement to make the manuscript clearer to readers.

Major concerns/comments:

  1. The authors have many figure panels (for example Fig. 1A, B) that are not referred to in the text. For Figs. 1-3, 6, it might make more sense to put all of the images as "Fig. #A" rather than labeling each image as a separate sub-panel.

Thank you for your suggestion. The Figure panels are now each referred to in text at least once. Figures 1, 2, and 3 have to remain separate due to figure size restrictions.

  1. The authors should include sub-panel labels (A, B, C, etc.) and relevant descriptions in the figure legends.

We have made the change as requested, subpanels are now referenced by label in relevant figure legends.

  1. In Fig. 1G, what does 0X and 2X refer to? This should be included in the figure legend.

This is correct, we removed “0X” and “2X”  and replaced with “vehic.” and “T4” respectively

  1. The figure legend for Fig. 1 appears incomplete (it ends in p>).

Figure legend is now properly visible.

  1. I don't understand the authors' statement "Staining of K14 was performed on sections to verify that the CPDPL promoter was indeed present in the corneal epithelium.". How does immunostaining for the K14 protein indicate the K14 promoter DNA element was present? Is the K14 not normally expressed in corneal epithelium, or is it unique to transgenic mice? I assumed the CPDPL expression was unique to the transgenic mice and expressed in the corneal epithelium, not the promoter that drives CPDPL expression.

Apologies for the confusion. We clarified the staining of K14 with the short paragraph: “In order to verify that CPDPL was present in the cornea of the CPDPL tg/wt mice, staining of K14 was performed on sections. This is because there are no readily available CPDPL antibodies and also to verify that the protein produced by the K14 gene was indeed present in the corneal epithelium. K14 staining gave bright signals at the limbus and clear signals in the basal corneal epithelium (Fig. 5). This suggests that K14 is indeed being expressed and the CPDPL gene placed downstream of the K14 gene should also be expressed.”

  1. On line 375, the authors state "However, p63a-expressing cells featured persistent CPD signal (fig. 7C highlighted 375 by arrows)". However, though the locations of these cells (indicated by arrows in Fig. 7C) are not indicated by arrows in Fig. 7D, they do not appear to stain positive for p63. Are these cells negative for p63? Perhaps the authors can provide arrows in both Figs. 7C and D to be more clear about which p63-expressing cells the authors are wishing to point out.

Thank you for noticing this issue. We changed 7C to 8C. The figure referenced in the text was the incorrect figure. We added white lines in both Figure 7 and 8 to better define the outlines of the limbus.

  1. In lines 491-493, the authors state that "the repair proteins would need to unwind the histones to access lesions, something unlikely to happen outside of cell division." First, the histones are not "unwound". Rather, they are displaced from the DNA to allow access of DNA repair proteins to UV photoproducts in DNA. Second, decades of research have shown that UV photoproducts can be removed from DNA in the absence of cell division, and thus the authors' statement that DNA repair proteins would not be able to access DNA outside of cell division is not correct.

Thank you for clarifying this, the paragraph incorrectly referring to histones has been removed.

  1. In figures 2, 3, and 6, the authors indicate the level of statistical significance as p>0.001 or 0.05. Do the authors mean to say less than (<)? If not, perhaps the authors should better clarify the comparisons that are being made here.

To clarify thie, we changed “p > 0.001” and “p > 0.05” to “p < 0.001” or “p < 0.05”

  1. In Figure 6, the authors should indicate what the arrows represent in the figure legend.

Thank you for noticing this issue, all arrows are now properly referenced in the figure legends.

  1. In Figure 9, is it possible for the authors to provide some type of quantitation showing the purported differences in CPD levels?

The newly added Figure 10 shows total number of CPD positive nuclei in the corneas subjected to the treatments shown in Figure 9. The figure shows statistical change in central CPD amount but no significant change in limbal CPD amount. This allows for quantitative comparison of CPDPL, T4N5, and hybrid treatments. Specialized equipment is under assembly to repeat the ex vivo experiments in vivo. This will likely be the subject of our next publication on the topic.

Minor comments:

  1. Line 79: It is called nucleotide excision repair, not nuclear excision repair.

We changed “nuclear” to “nucleotide”

  1. Line 158: The abbreviation HLE should be defined the first time it is used.

Thank you for noticing this, HLE is defined in both the methodology 2.1 title “Primary human limbal epithelial (HLE) cell culture” and in methodology 2.2 “Human limbal epithelial (HLE) cells were seeded…”

  1. Line 158: the 4 should be superscript

The 4 is now superscript

  1. In several places in the manuscript (including line 158), the 2 in "cm2" should be superscript.

We changed every instance of “cm2” to “cm2

  1. Line 174: x g, not G

Changed “G” to “×g”

  1. Line 214: the word diluted is repeated

Removed repeated word

  1. Line 223: mto?

Changed „mto“ to „to“

  1. Line 349: were, not where

Changed „where“ to „were“

  1. Line 376: containing is misspelled

Apologies, spelling mistake has been corrected

  1. Line 378: there are two periods at the end of the sentence.

Excess period removed

  1. Line 483: insert ref should be fixed

Inserted correct references

  1. The abbreviation CPDPL should be defined somewhere in the manuscript. Presumably it means CPD photolyase.

Defined the abbreviation in the abstract and the introduction with “The DNA repair enzymes CPD photolyase (CPDPL)…”

  1. In some cases, the authors capitalize the 'F' in "Fig. x", but in others they do not. The authors should be consistent, and preferably capitalize it.

Capitalized all instances of “fig” to “Fig”
